# Determination of Soil Electrical Conductivity and Moisture on Different Soil Layers Using Electromagnetic Techniques in Irrigated Arid Environments in South Africa

**Phathutshedzo Eugene Ratshiedana** [1,2,*], **Mohamed A. M. Abd Elbasit** [3], **Elhadi Adam** [2], **Johannes George Chirima** [1,4], **Gang Liu** [5] **and Eric Benjamin Economon** [1]

1   Agricultural Research Council-Natural Resources and Engineering-South Africa, 600 Belvedere Street, Arcadia, Pretoria 0083, South Africa
2   School of Geography, Archaeology and Environmental Studies, University of the Witwatersrand, Johannesburg Private Bag x3, Wits, Johannesburg 2050, South Africa
3   Department of Physical and Earth Sciences, School of Natural and Applied Sciences, Sol Plaatje University, Kimberley 8300, South Africa
4   Department of Geography, Geoinformatics and Meteorology, University of Pretoria, Pretoria 0028, South Africa
5   Institute of Soil and Water Conservation, Chinese Academy of Sciences and Ministry of Water Resources, Xianyang 712100, China
*   Correspondence: ratshiedanap@arc.agric.za; Tel.: +27-012-310-2598

**Abstract:** Precise adjustments of farm management activities, such as irrigation and soil treatment according to site-specific conditions, are crucial. With advances in smart agriculture and sensors, it is possible to reduce the cost of water and soil treatment inputs but still realize optimal yields and high-profit returns. However, achieving precise application requirements cannot be efficiently practiced with spatially disjointed information. This study assessed the potential of using an electromagnetic induction device (EM38-MK) to cover this gap. An EM38-MK was used to measure soil apparent electrical conductivity (*ECa*) as a covariate to determine soil salinity status and soil water content θ post irrigation at four depth layers (Hz: 0–0.25 m; Hz: 0–0.75 m; Vz: 0.50–1 m). The inverse distance weighting method was used to generate the spatial distribution thematic layers of electrical conductivity. The statistical measures showed an $R^2 = 0.87$; r > 0.7 and $p \leq 0.05$ on correlation of *ECa* and SWC. Based on the South African salinity class of soils, the area was not saline *ECa* < 200 mS/m. The EM38-MK can be used to estimate soil salinity and SWC variability using *ECa* as a proxy, allowing precise estimations with depths and in space. These findings provide key information that can aid in irrigation scheduling and soil management.

**Keywords:** apparent electrical conductivity; salinity; soil moisture; em38-mk; inverse distance weighting; spatial distribution

## 1. Introduction

Soil moisture is a widely perceived component of the water balance, crucial for crop development and nutrient transport from the soil to crops [1,2]. Apart from its role in agriculture, soil moisture plays an important role in the Earth's energy balance by controlling the changes of surface water and energy fluxes to balance atmospheric water demands through evaporation [3]. Generally, soil moisture in arid environments is a limited component of the hydrological cycle due to limited and variable rainfall and high temperatures, along with long spells of droughts and lack of water sources [4]. However, most arid regions contain arable land, and usually the supplemental irrigation that they rely on consists of water transfers from areas with ample availability [5].

Given the challenges faced by arid areas, and with the scarcity of water, the need to monitor, manage, and control water application rates through irrigation scheduling

remains a key mitigation approach [6]. That being stressed, various factors come into play when irrigation events are scheduled; one of the most important components of accurately determining prior irrigation scheduling is soil moisture. Accurate measurement and estimations of soil moisture allow farmers to fully understand the water dynamics in their cropped fields, both spatially and with depth [7]. The spatial distribution of soil moisture in water-scarce South Africa is a major challenge, particularly within the agricultural sector where moisture information is critical for irrigation scheduling. Most measurements are based on point data, which lack the spatial representativeness of the area. With improvements in smart agriculture, point information does not provide much guidance on spot treatment of soil or on scheduling irrigation only in needy areas.

Previous studies have demonstrated various methods of monitoring soil moisture from plant level, field scale, and global scales to assist farmers in proper irrigation scheduling [8,9]. Several strategies to measure and monitor soil moisture have been implemented, including the application of gravimetric methods where soil is sampled, weighed, dried, and weighed again to calculate the water loss. Apart from the above methods, others such as nuclear-based soil moisture assessment, tensiometric approaches, and hygrometric approaches have been widely explored [10–12]. However, all soil moisture measuring and estimation methods define moisture from a fixed point perspective without taking into account the spatial variability of soils within the field [13]. Furthermore, methods such as gravimetric approaches come with some limitations, such as induced errors on measuring scales, human errors when recording data, and high cost [14,15].

The use of probes, such as the capacitance probes, measuring the dielectric current to estimate moisture at a point level also have challenges such as inducing errors, particularly in clay soils where mud cracks are more common during moisture deficit periods. Such cracks create an air space, removing the sensor contact to the soil, which results in null readings occurring. This is not only the case with capacitance probes installed in fields; it applies to any contact sensors that may be used to obtain instant readings [16,17]. Although such approaches provide more detailed information about soil water content depths and salinity status, they also have spatial limitations [18]. Despite the fact that point measurements lack spatial representation of an area, they are still crucial for calibration and evaluation of spatial estimation approaches [19].

With increasing interest in precision farming and smart agriculture, soil properties such as soil electrical conductivity (EC), texture, permeability, and soil moisture are some of the most crucial, which farmers and agricultural advisors need to understand prior to any land use [20,21]. Spatial data on soil salinity and moisture availability carry crucial information, which aids in farm irrigation management and soil treatment due to the heterogeneity of soil properties in space and time [22]. The transformation from traditional point soil moisture and EC measurement, through destructive soil sampling, to geospatial approaches has gained more attention in attempts to solve water scarcity problems while increasing water use efficiency and resource sustainability [23,24].

The development of satellite-based remote-sensing soil moisture products has added value in understanding soil water dynamics spatially. Global soil moisture products have been developed from satellite remote sensing and are widely used in large-scale investigations. These products include the Soil Moisture Active Passive (SMAP) at 36 km with a daily temporal resolution retrieval; Soil Moisture and Ocean Salinity (SMOS) at 25 km and daily retrieval; and Scanning Multichannel Microwave Radiometer (SMMR) at 150 km on daily basis, to list just a few [25,26]. However, due to their coarse resolutions, their applicability at farm scales is challenging. With such challenges, various researchers have integrated different satellite data, such as the Moderate Resolution Imaging Spectroradiometer (MODIS) with 250 m resolution [27,28], Landsat 8 OLI at 30 m resolution [29], and Sentinel-1 at 10–20 m resolution, to aid in understanding moisture dynamics at field scales [30]. Despite these advances, soil moisture spatial data obtained from remotely sensed observations only estimate soil moisture within the top 0–5 cm; this already is a limitation when assessing moisture dynamics for deep-rooted crops [31].

The limited coverage of soil depth and low spatial resolution of these datasets do not adequately address many issues at national scale and, more importantly, at local field scales. The wide range of image coverage and ease of access enables their applications from regional to global scales, however, mixed pixels are common in remote sensing data, making monitoring soil moisture variability in heterogeneous terrains more difficult [32]. With such challenges in the past decade, advances and innovations in science and technology have led scientists to develop sensors that can estimate soil moisture indirectly, with reasonable acquisition depths such that the root zone of various crops can be reached and their information can be integrated in geospatial tools to develop spatial digital soil moisture maps [33–35]. Some of the most promising sensors include the non-destructive electromagnetic induction sensors and ground penetration radar geophysical tools used in groundwater exploration [36,37].

However, such sensors also do not directly measure volumetric water content but rather measure a different parameter, which can be related to soil water content as a surrogate [38,39]. Unfortunately, apart from the use of hyperspectral sensors, until now there have been no satellite sensors providing information on soil parameters such as EC, soil types, and soil fertility, which are important properties of the soil that play a crucial role in water absorption by plants [40]. Consequently, spatial information of soil EC remains the biggest data gap challenge. In areas with high salinity problems, EC plays a critical role in delineating zones for salinity treatment. The lack of spatial data on soil EC requires ground measurement of soil EC or laboratory analysis of field sampled soil, which is a tedious exercise in larger areas. High salinity zones are more prone to water logging and poor crop performance.

Devices such as electromagnetic induction sensors (EM38) have indirectly come a long way in the history of soil moisture estimation by measuring apparent electrical conductivity (*ECa*) and relating it to soil water content through calibration methods [41–48]. The EM38 is a non-destructive soil sensor, which retrieves soil electrical conductivity as a function of soil moisture availability to move the charges available and conduct electrical current flow [47]. The sensor takes measurements to a depth of 1 m. The EM38 provides high temporal resolution data, at as high as 1 s intervals, however, EM38 sensors are expensive. The use of EM38 sensors for salinity assessment used to be a time-consuming activity prior to the introduction of smart sensors and precision technology. Currently, EM38 devices can be towed in farm machineries and protected with non-metallic covers to quickly survey an entire field in a short space of time. Its ability to estimate soil properties at four soil depths makes it attractive and necessary for understanding soil moisture dynamics at different root zones, especially in irrigated fields where scheduling irrigation is a prerequisite.

Salinity and soil moisture content availability are the two major concerns in irrigated farming regarding soil quality and water quantity. However, the availability of spatial soil water content information in the South African farming environment is very limited. As such, point information dominates most areas. This information provides crucial guidance in agricultural planning and farm management in order to solve water and treatment loses. EC can be used as a surrogate for soil water content estimation in situations where moisture sensors are not available to provide the spatial representativeness of an area. This study aims at mapping the spatial (Hz) and vertical (Vz) distribution of soil electrical conductivity and soil moisture at different soil depths by integrating electromagnetic data with geospatial techniques under different wetness conditions post irrigation.

## 2. Materials and Methods

### 2.1. Study Area Description

The study was carried out on an 18 ha experimental farm owned by the South African Barley Breeding Institute (Figure 1a). The area is located 12 km from Hartswater town and is part of the Phokwane Local Municipality in the Frances Baard District Municipality (Figure 1c) of the Northern Cape Province (Figure 1b). Geographically, the 18 ha experimental farm is located between the coordinates 27°43′14.41″ S and

24°44′34.18″ E. The experimental farm is located within the Vaalharts irrigation and covers an area of over 36,950 hectares, being the largest irrigation scheme in South Africa. Ojo et al. [49] reported that the Vaalharts Irrigation Scheme was planned in 1933 to address unemployment, hunger, and poverty. The proposal to develop this scheme was approved in 1934, which resulted in the construction of the Vaal dam. Water was then diverted from the Vaal River weir on the eastern side of Warrenton [50]. As the water table rose, the scheme's salinity problems initiated and worsened waterlogging, which to date prevents the use of groundwater in the area due to salt accumulations [50]. The farm is situated between 1088 and 1090 m above sea level.

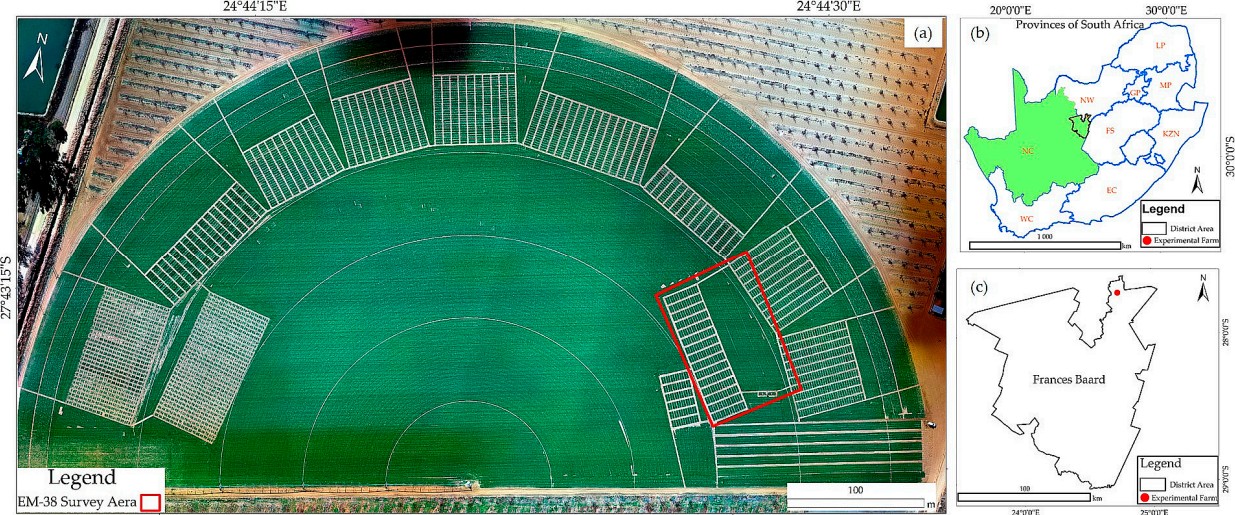

**Figure 1.** Locality map of the study area showing the experimental farm and the area of interest used for the EM38 survey where (**a**) is the 18 ha experimental farm, the red boundary is the EM38 surveyed area, (**b**) shows the provincial layout of South Africa, highlighting Northern Cape (NC) where the study area is located in green, while (**c**) shows the location of the survey area in the district municipality.

According to Vermeulen and van Niekerk [51], sandy soils and inadequate natural drainage in Vaalharts are well known for causing waterlogging. The months of December and January typically have the highest monthly average temperature, of approximately 32 °C, while July has the lowest monthly average temperature of 0.5 °C. The summer season, which runs from October to April, has an average precipitation of 450 mm [52]. The area receives summer rainfall with hot days, and has a cold winter season [53]. The Vaalharts irrigation scheme is in operation for one of the major irrigated pecan producers, and it uses pivot irrigation systems [54]. The fact that the area is the biggest pivot-irrigation-dominant scheme in South Africa and located in a dry area made it attractive for the undertaking of this study for crop water use assessment, especially with the fact that agriculture dominates the use of freshwater in the country [54].

### 2.2. Methods and Approaches

2.2.1. Soil Electrical Conductivity Surveying

Four field survey campaigns were conducted within a selected area, located in an experimental barley crop field, using an electromagnetic induction EM38-mk device (Figure 2). The device is equipped with a signal transmitting coil at one end and a receiver coil on the other end, spaced 1 m apart.

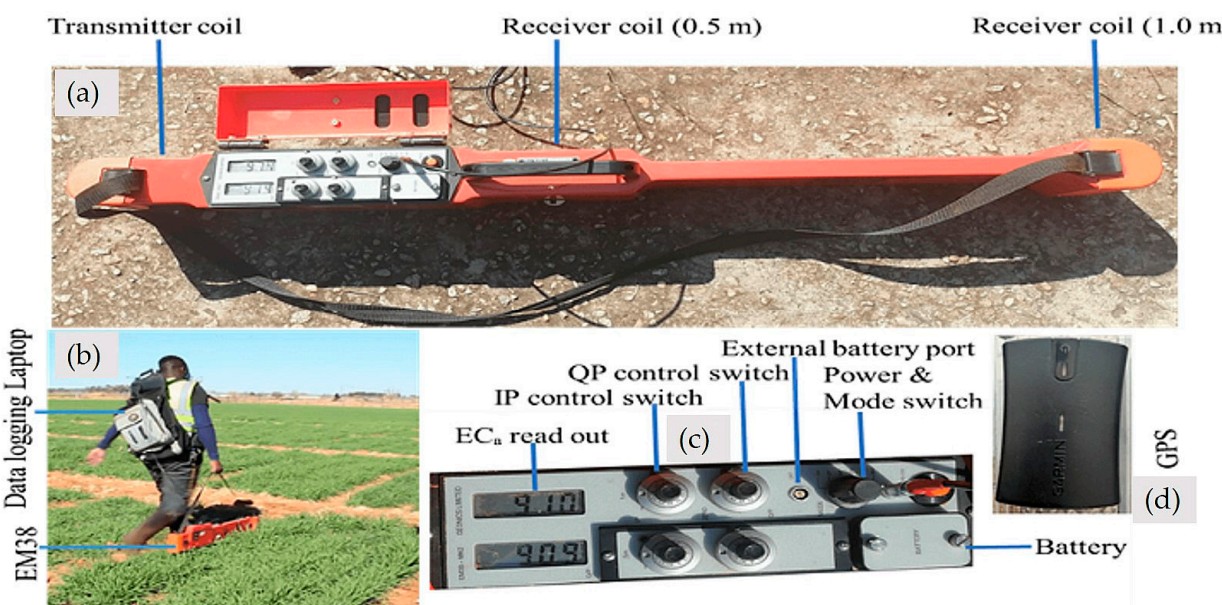

**Figure 2.** An electromagnetic induction instrument (EM38MK) and its components, where (**a**) is the EM38 device, (**b**) is the device on vertical mode during survey, (**c**) shows the setting controls, and (**d**) is the Bluetooth GPS linked to the device for sample location.

The device was used to survey the same area twice on each field campaign to allow for two survey orientations wherein one survey was done with the instrument on vertical orientation, and another on horizontal orientation. The soil signal penetration depth when the device is on vertical mode (Vz) is 0–50 cm and 0–100 cm, and when the device is on its horizontal orientation (Hz) the measurement depth is 0–25 cm and 0–75 cm. The same height of 10 cm above the soil was maintained using belt straps, reducing the seesaw effect on the instrument. Before the start of each survey, the battery status of the device was checked. The EM38 device was calibrated at 1.5 m height above ground each time before the survey was done, according to the Geonics® instructions published in Canada. The surveyed area was cleared of all metallic objects to avoid the introduction of artificial magnetic noise, which included the removal of metal belts, watches, cell phones, and any electronic devices that could affect the measurements. The survey lines followed the raw paths available between the barley crops, which changed with different field campaigns as the crop grew and canopy cover became dense. All stations surveyed were located using a GPS connected to the computer, and the computer was connected to the instrument for data logging during the survey (Figure 2).

### 2.2.2. Calibration of the Electromagnetic Induction Instrument (EM38)

To calibrate the electromagnetic induction instrument (EM38), apparent soil electrical conductivity was measured in the horizontal mode (0.25–0.75 m) and vertical mode (0.50–1 m) within the EM38 survey area of interest. Following the EM38, soil samples were collected at four different sites within the EM38 survey area of interest to determine soil properties. At each point, a hand auger was used to take cores from 0–1 m at an interval of 0.25 m with depth added to four cores being taken from each point (Figure 3a). For every core taken, physical properties including in situ soil water content, electrical conductivity, and temperature were measured before sample packaging.

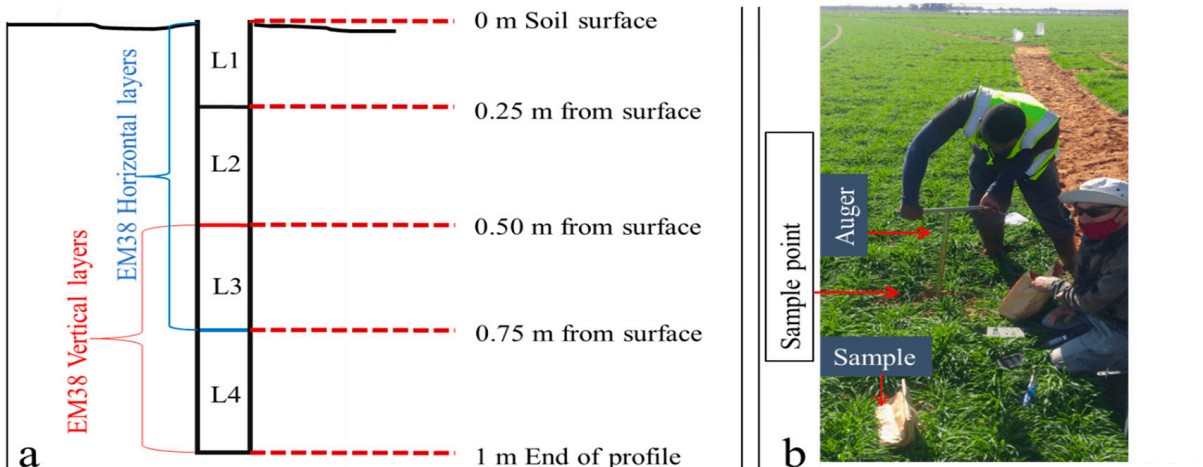

**Figure 3.** Soil sampling at different depths for EM38 calibration. (**a**) Different layers of soil taken at each calibration point, (**b**) surveyors taking core samples at different depths equivalent to the EM38 measurement depth.

All core soil samples were stored in zip lock bags labelled with GPS coordinates, depth, and sample number; zip lock bags were used to prevent moisture loss from the samples (Figure 3b). The samples were taken to the laboratory for physiochemical analysis. Similar soil properties, including soil water content and electrical conductivity saturated at 1:2.5, were analyzed at the laboratory. Additional soil properties that included the SAR properties (Mg, Ca, Na, and K), cations, and the soil pH were also analyzed as other soil attributes influencing the *ECa* response. Linear regression analyses were applied for all depths and sample points to establish a relationship between temperature corrected *ECa*-EM38, *ECa* from EC-meter, ECe, soil water content, and other soil properties, which best describes their correlations.

2.2.3. Spatial Distribution of Soil Electrical Conductivity

EM38 readings were used to estimate the spatial distribution patterns of soil electrical conductivity during various irrigation events. The survey's EM38 measurements were interpolated in ArcGIS 10.6 Esri© software using an inverse distance weighting interpolation to map the spatial distribution of electrical conductivity within the barley. The inverse distance weighting interpolation approach was selected due to its ability to interpolate unknown areas without overestimating or underestimating electrical conductivity measurement readings [55]. The maps depicting the apparent electrical conductivity of the soil were produced with a pixel size of 35 cm × 40 cm. For all four field campaigns, spatial distribution maps were generated for four depths of farm soil at the selected area of interest, with the first layer representing the root zone of the barley crop. All apparent electrical conductivity readings from the EM38 were inspected for out of range, including negative and zero, readings. Electrical conductivity readings from each survey station were corrected for temperature effects using [56] Equation (1) given as:

$$ECa(T25) = ECa[0.4479 + 1.3801e^{\frac{T}{26.815}}] \tag{1}$$

where *ECa* is the apparent conductivity measured at a point and T is the mean daily temperature with depth 0–1 m obtained from Dirk Friedhelm Mercker (DFM) probes located in the field.

2.2.4. Assessment of Soil Salinity

Soil salinity within the study area was assessed based on the salinity, alkalinity, and sodicity of soils in South Africa listed in Table 1 [57].

**Table 1.** Salinity, alkalinity, and sodicity of South African soils.

| Soil Class | EC Concentration (mS/m) |
|---|---|
| Non-Saline | <200 |
| Slightly Saline | 200−400 |
| Moderately Saline | 800−1600 |
| Strongly Saline | >1600 |
| Saline-Sodic (Non-Alkaline) | >400 pH < 8.5 |
| Saline-Sodic (Alkaline) | >400 with pH > 8.5 |
| Sodic | <400 with pH > 8.5 |

2.2.5. Conversion from *ECa* to SWC

From the interpolated electrical conductivity spatial distribution layers, soil water content was estimated by establishing a calibration relationship between in situ soil water content and soil electrical conductivity. A power regression model was developed from EM38 field campaign data to estimate the soil water content percentage in four layers of each campaign. In the developed calibration equation, electrical conductivity was set as the independent variable, while soil water content was selected as a dependent variable. The relationship between soil water content (θ) and soil electrical conductivity (*ECa*) was expressed as:

$$\theta = ECa^B \tag{2}$$

where θ represents the soil water content, and *ECa* is the apparent electrical conductivity; to determine θ, Equation (2) was applied to the *ECa* raster files in the raster calculator in the ArcMap environment to produce θ maps.

2.2.6. Data Processing and Evaluation Approach

The interpolation approach was used to process and develop the spatial distribution maps of soil electrical conductivity, while the statistical approach for evaluating the relationships between the apparent electrical conductivity (*ECa*), soil water content (θ), and other soil attributes were used. The relationships were established using regression analysis. A correlation coefficient (r) with a coefficient of determination ($R^2$) values was established in every relationship between soil electrical conductivity and soil attributes, as the primary measure or model of quality of fit. Furthermore, correlation analysis using Pearson's correlation (*p*-value) was conducted to understand the relationship between soil water content (θ) and the soil electrical conductivity, in order to assess whether the relationship had a statistically significant difference.

**3. Results**

*3.1. Assessment of Soil Electrical Conductivity to Determine Soil Water Content*

3.1.1. Spatial Distribution of Soil Electrical Conductivity

The mapping of apparent soil electrical conductivity was conducted at a selected area at a farm scale using an electromagnetic induction instrument (EM38-MK), integrated data with an geospatial interpolation inverse distance weighting method, and ground-based sampling approaches with a focus on using electrical conductivity in determining soil water content at different depths. The spatial variability maps of soil electrical conductivity within the study area are shown in Figures 4–7. Variability in the soil's electrical conductivity within the selected area varied both spatially and with depth. To minimize sampling of soil and measuring of soil water content manually at each depth during each campaignfrom the 22 September 2020 EM38 field campaign, to the last EM38 field campaign on 14 October 2020 (Figures 4–7)—soil water content measurements were only conducted on the first top layer (0–25 cm). These measurements were used to quantify the variability of the spatial soil water content on other layers by calibration from the first day of the EM38 campaigns.

Higher values were obtained from 0.75 m to 1 m depths, while the upper layers between 0 and 0.50 m had low values. The low values obtained could possibly be related to the root zone water abstraction by the crops and possible rapid soil water evaporation as a result of loose soils due to tillage on the near surface layers, characterized as sandy-loam as described in the study area description. The bottom layers with high conductivity can be related to high water content, which slowly responds to the near surface activities, including root zone water extraction by roots and top surface atmospheric demand. In August, the EM38 *ECa* response values ranged between 1.1 mS/m and 11.85 mS/m, with an average soil water content of 6.63%. On 22 September 2020, the *ECa* values ranged between 5.1 mS/m and 19 mS/m at a 16.26% soil water content average. The results from 23 September 2020 show that the *ECa* readings ranged from 7.7 mS/m to 22.4 mS/m at an average soil water content of 18.77%, while predictions of the last campaign on 14 October 2020 showed *ECa* ranging from 1.1 mS/m to 18.6 mS/m, with 5.33% average soil water content.

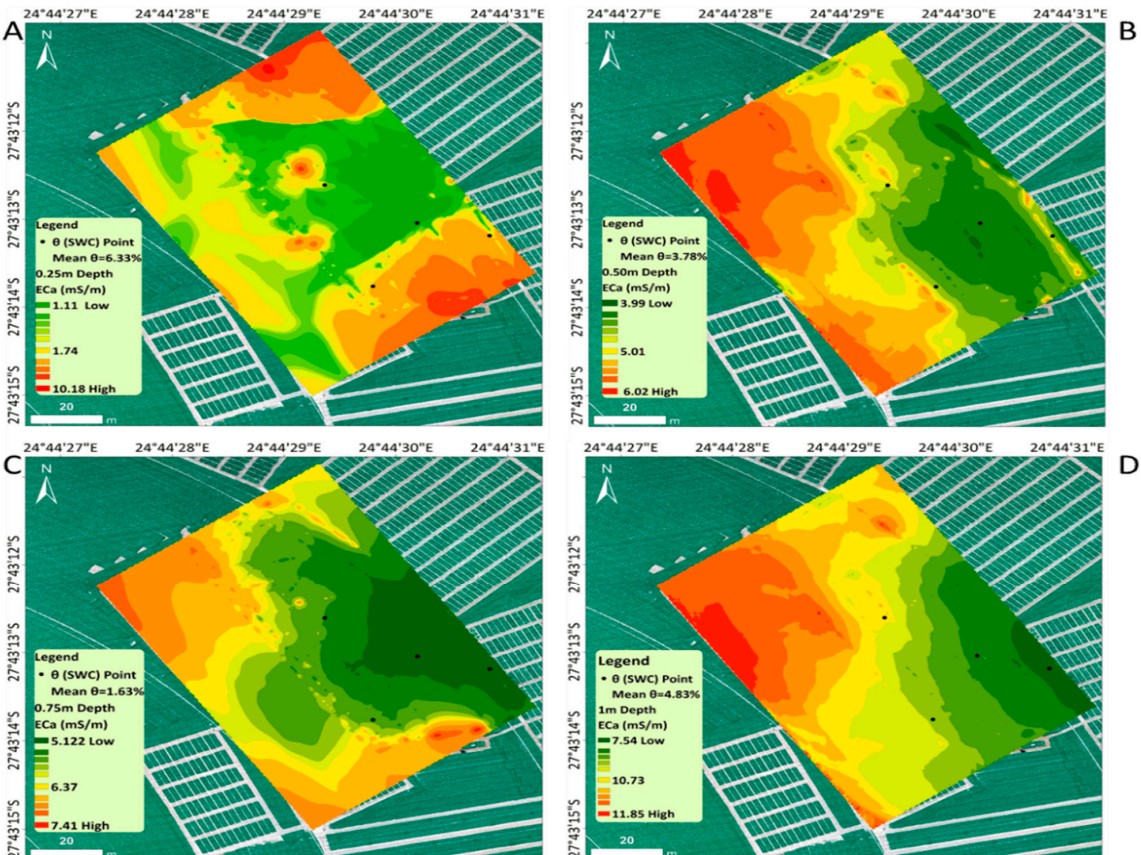

**Figure 4.** Spatial variability of soil electrical conductivity for four depths ((**A**–**D**) represent top to bottom layers on 11 August 2020).

The soil electrical conductivity shown in Figure 4 clearly shows a variability pattern of *ECa* at four different layers between 0 m and 1 m depths. It is clear that the *ECa* at each layer differs from that of other layers; this is due to the fact that *ECa* and any other soil property vary spatially no matter how small the area is. It is also evident that the range of *ECa* during this survey period was between 1.1 and 11.85 mS/m, which can be linked to soil wetness or dryness. Low *ECa* in this context can be a sign of low soil water content, as described in the works of Bai et al. [57] and Turkeltaub et al. [58].

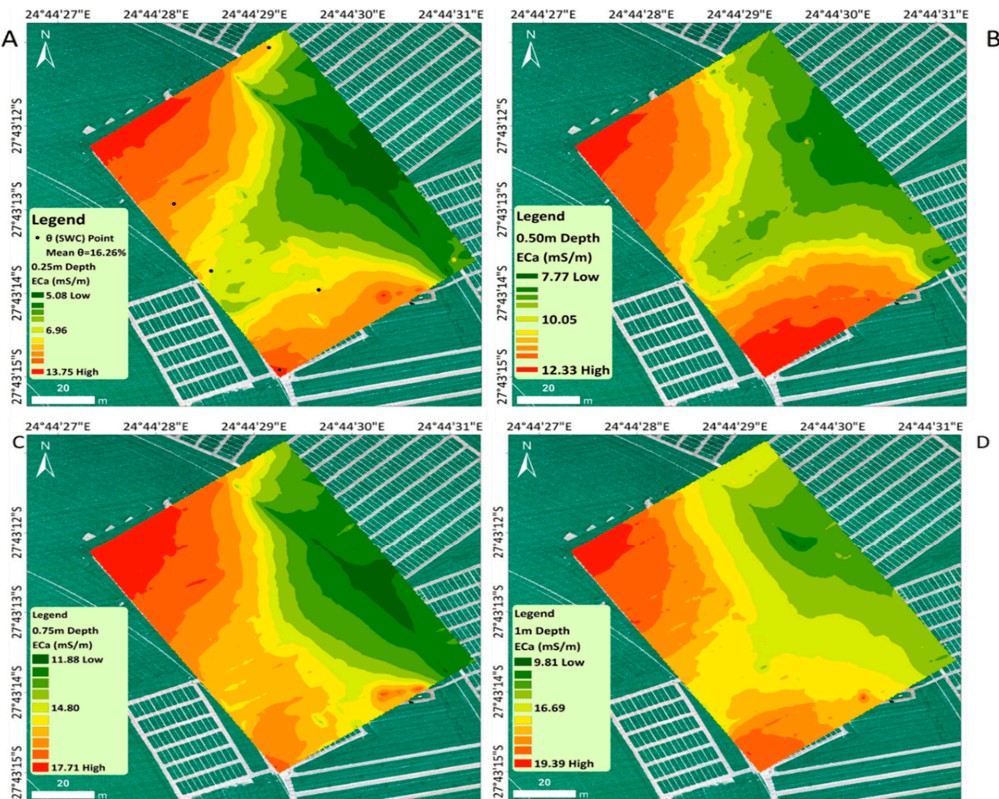

**Figure 5.** Spatial variability of soil electrical conductivity for four depths ((**A**–**D**) represent top to bottom layers on 22 September 2020).

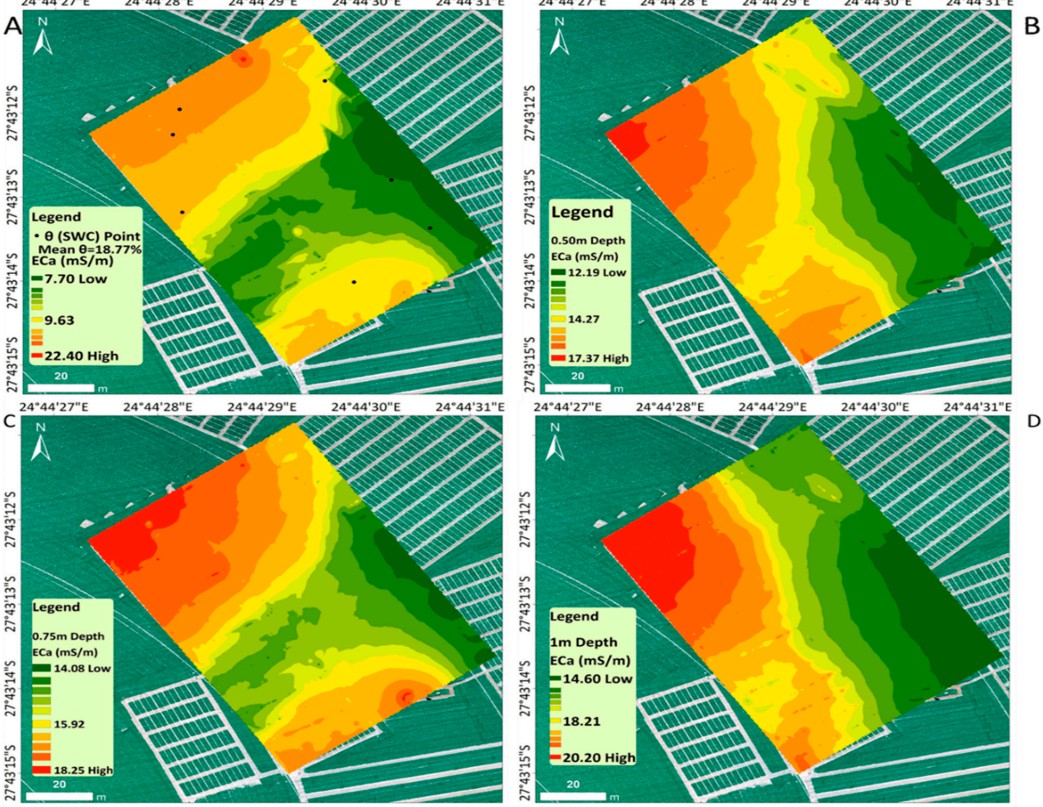

**Figure 6.** Spatial variability of soil electrical conductivity for four depths ((**A**–**D**) represent top to bottom layers on 23 September 2020).

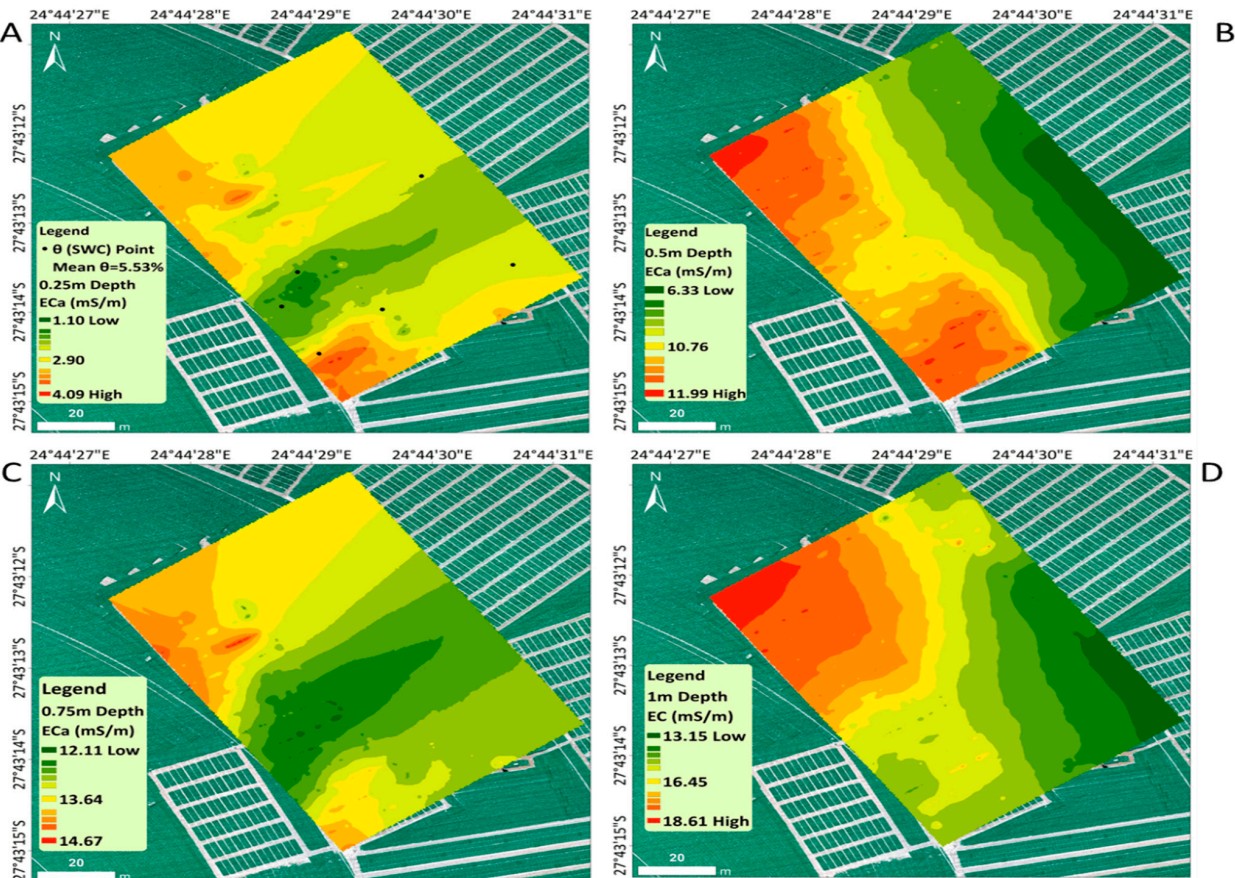

**Figure 7.** Spatial variability of soil electrical conductivity for four depths ((**A**–**D**) represent top to bottom layers on 14 October 2020).

Figure 5 indicates the spatial distribution of soil *ECa* for four depths in the surveyed area. It is clear that higher *ECa* values are more pronounced on the western part of the map, cutting out almost half of the surveyed area, while the eastern part has slightly low values, which are higher compared to the distribution indicated in Figure 4. This is an indication of high soil water content allowing more conductivity in soils. The *ECa* in this case ranges between 5 and 19.4 mS/m. On the map Figure 4A the mean SWC = 6.33%, as compared to the Figure 5 where the mean SWC = 16.26%. It is evident that this has increased, while the *ECa* values also increased with maximum *ECa* = 13.75 mS/m.

Figure 6 depicts an increase in soil *ECa* across all depths of layers. The *ECa* ranges between 7.7 and 20.3 mS/m. A higher *ECa* is evident in the northern and southern parts of the surveyed area, while it appears to be lower in the central parts of the 0–25 m upper layer and 0.75 m, while in 0.50 and 1 m *ECa* appears to be higher in the western part of the surveyed area. The higher *ECa* might be attributed to high soil water content allowing high conductivity in soils. This is due to the fact that the mean θ% continues to increase when compared with Figure 5, where mean θ is 16.26%, and Figure 6, where it is 18.77%, while the *ECa* values on the maps also increase. The relationship where *ECa* increases with SWC has been narrated by Bai et al. [57].

Figure 7 shows a slight decline in soil *ECa*, which might be attributed to a decrease in soil water content, limiting soils to conducting more current as opposed to wet soils. It is evident across all layers that soil *ECa* varies with space, with the lower layer containing higher *ECa* values compared to the rest of the layers.

### 3.1.2. Relationship between In Situ Soil Electrical Conductivity and In Situ Soil Water Content

Prior to the conversion of the soil's electrical conductivity into soil water content, it was important to clearly understand the relationship between the measured soil electrical conductivity and soil water content from the data obtained during the first day calibration sampling with depth. The relationships between the in situ soil electrical conductivity, taken using EC meter, and soil water content measured using amplitude domain reflectometry soil moisture (ADR), probe at different soil depths and different soil sampling points (Figure 8a–d). The results show that soil electrical conductivity is generally influenced by soil water content availability. For this reason, low conductivity corresponds with low soil water content, while the opposite is the case with high values of soil water content and electrical conductivity. This is evident in Figure 8a–c while Figure 8c shows very low correlation, which might have been attributed to instrument measurement errors.

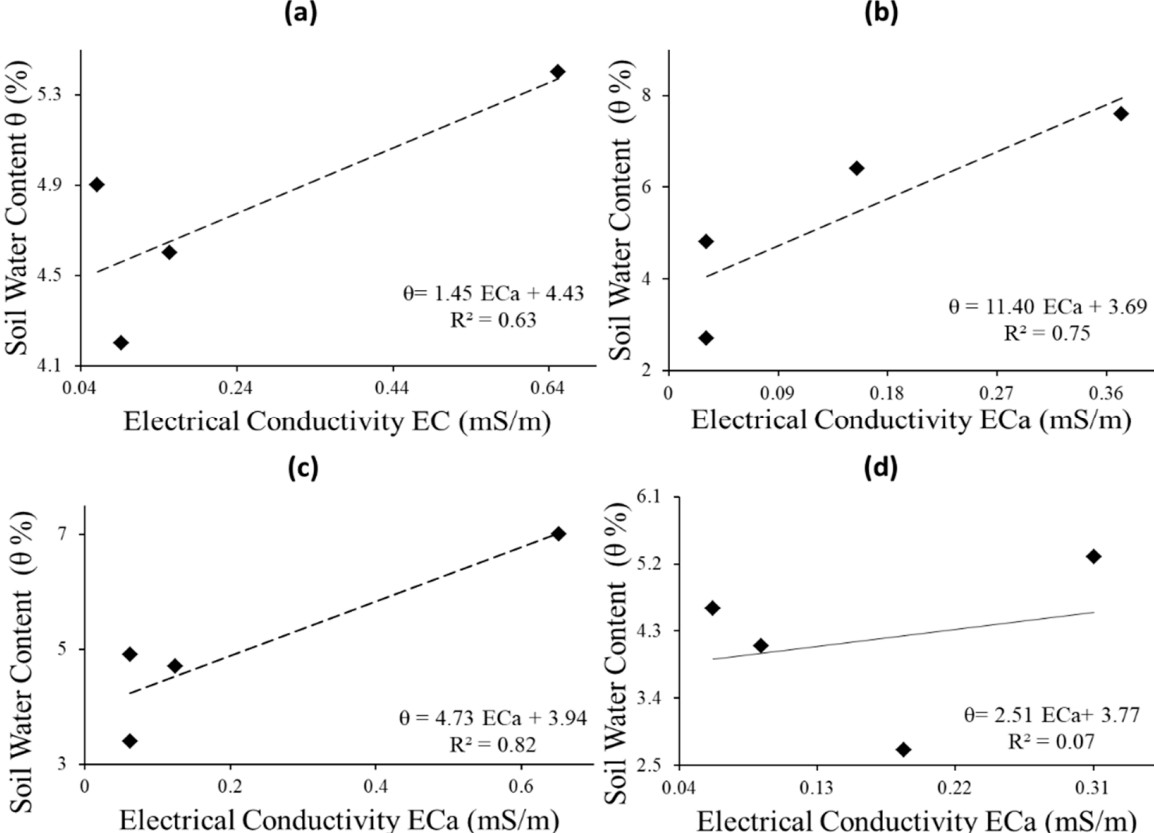

**Figure 8.** Relationship between in situ soil electrical conductivity and soil water content at varying soil depths at sampled locations, (**a**) represent the relationship of *ECa* and Soil water content at 0 to 0.25 m depth, (**b**) 0.25 to 0.50 m depth, (**c**) 0.50 to 0.75 m depth and (**d**) denotes a relationship at 0.75 to 1 m depth.

### 3.1.3. Soil Electrical Conductivity Conversion to Soil Water Content

Once the relationship between soil water content and electrical conductivity was understood, the findings were combined into one general model. Soil water content spatial distribution maps were generated by applying the developed model from the relationship of soil water content with the soil electrical conductivity to spatial distribution maps (Figure 9).

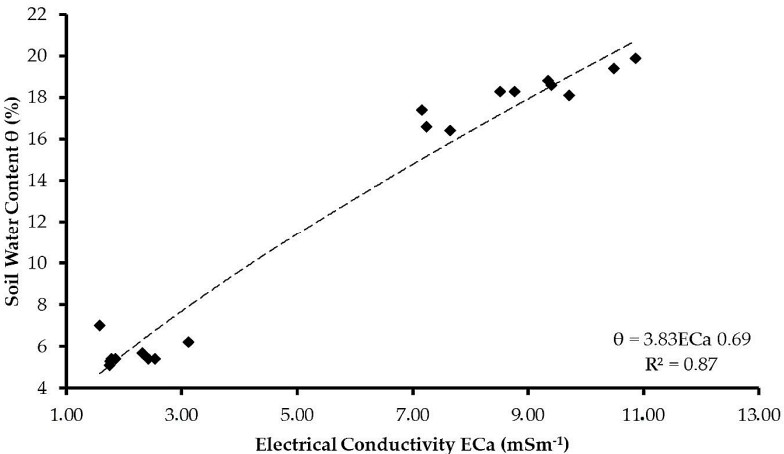

**Figure 9.** Model used to convert *ECa* to soil water content (θ).

### 3.2. The Spatial Distribution of Soil Water Content

The predicted spatial variability maps of soil water content determined based on the *ECa*-SWC relationship are given in Figures 10–13, and soil water content is given as a percentage of wetness in the soil. Soil water content was found to be higher in the bottom two layers, located at 0.75 m and 1 m depths. This is due to a high *ECa* shown on the spatial distribution maps of *ECa* for different survey days. On 8 August 2020, the predicted soil water content ranged between 4.15% in the top layer and increased up to 27.82% in the bottom layer. The amount of soil water content was lower in the first layer and increased with depth (Figures 10–13). The soil water content predicted on 22 September 2020 ranged between 13.85 and 39.85 percent, and soil water content increased with an increase in depth.

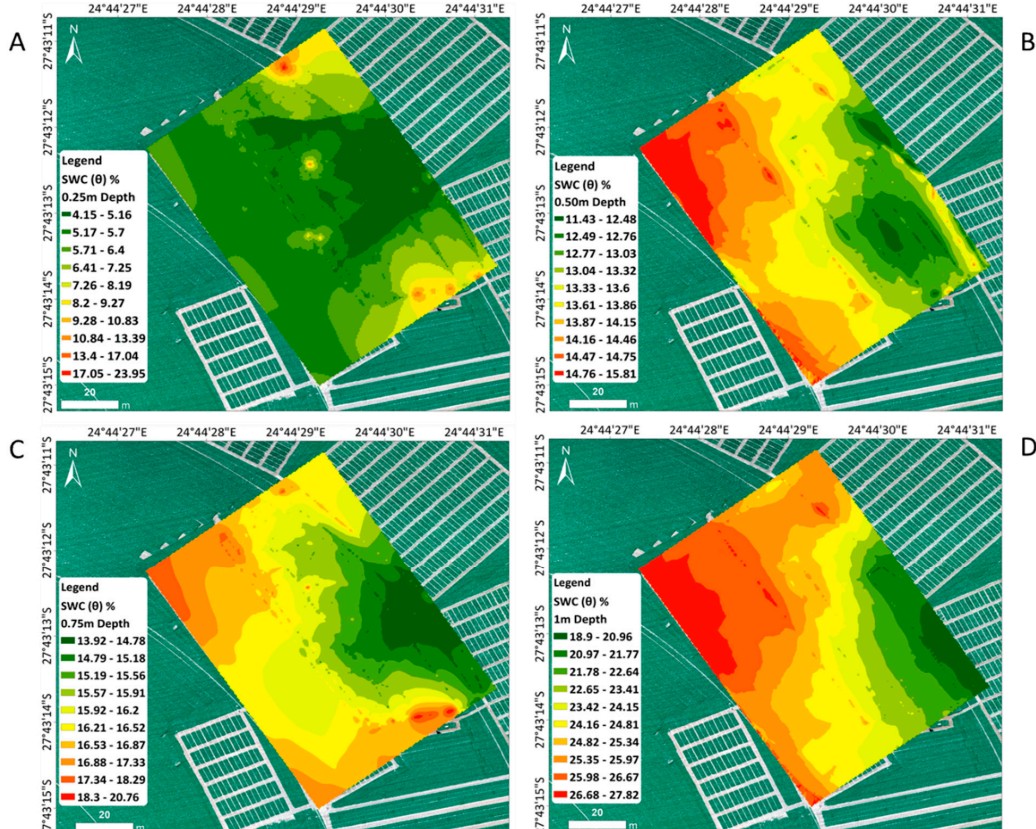

**Figure 10.** Predicted θ at four depths ((**A**–**D**) represents 0.25 m to 1 m on 8 August 2020).

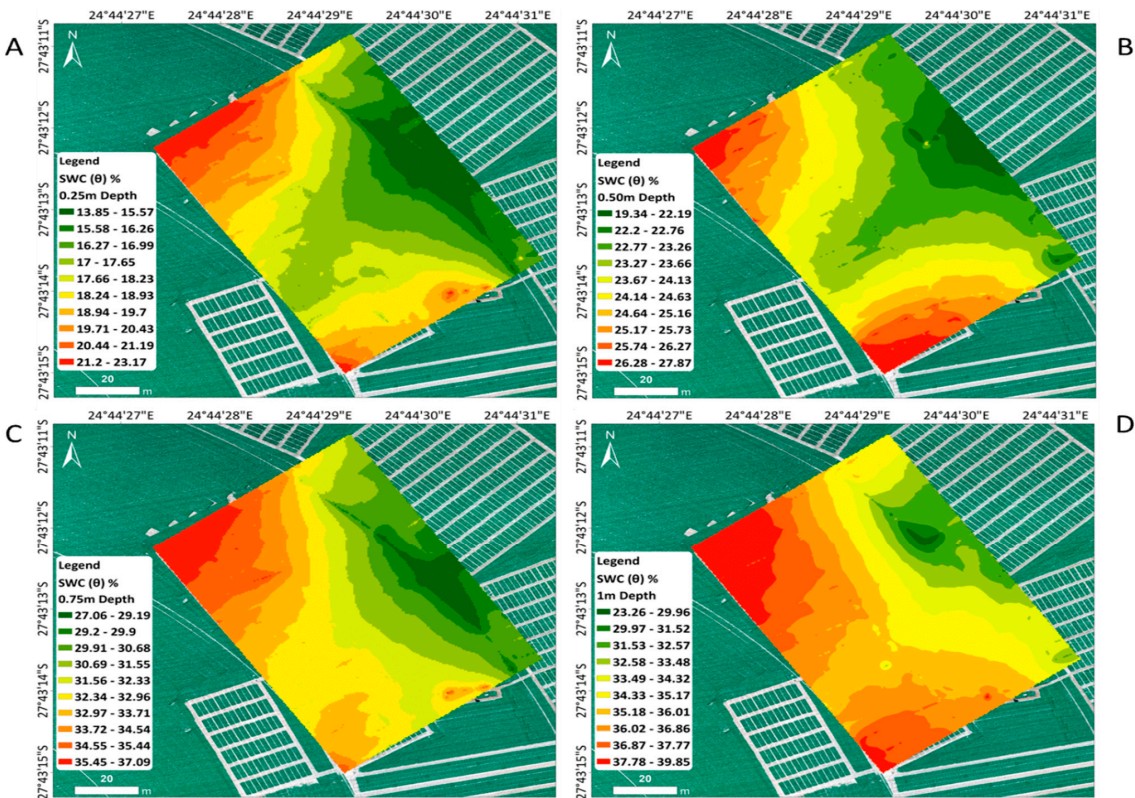

**Figure 11.** Predicted θ at four depths ((**A–D**) represents 0.25 m to 1 m on 22 September 2020).

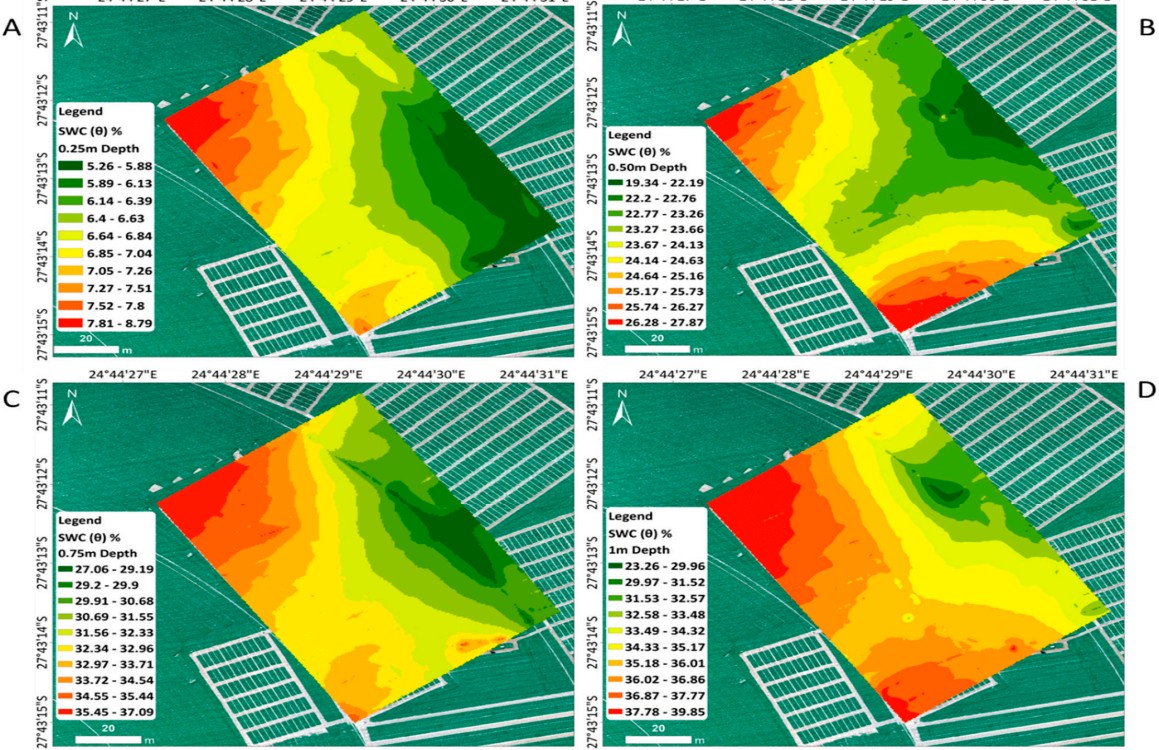

**Figure 12.** Predicted θ at four depths ((**A–D**) represents 0.25 m to 1 m on 23 September 2020).

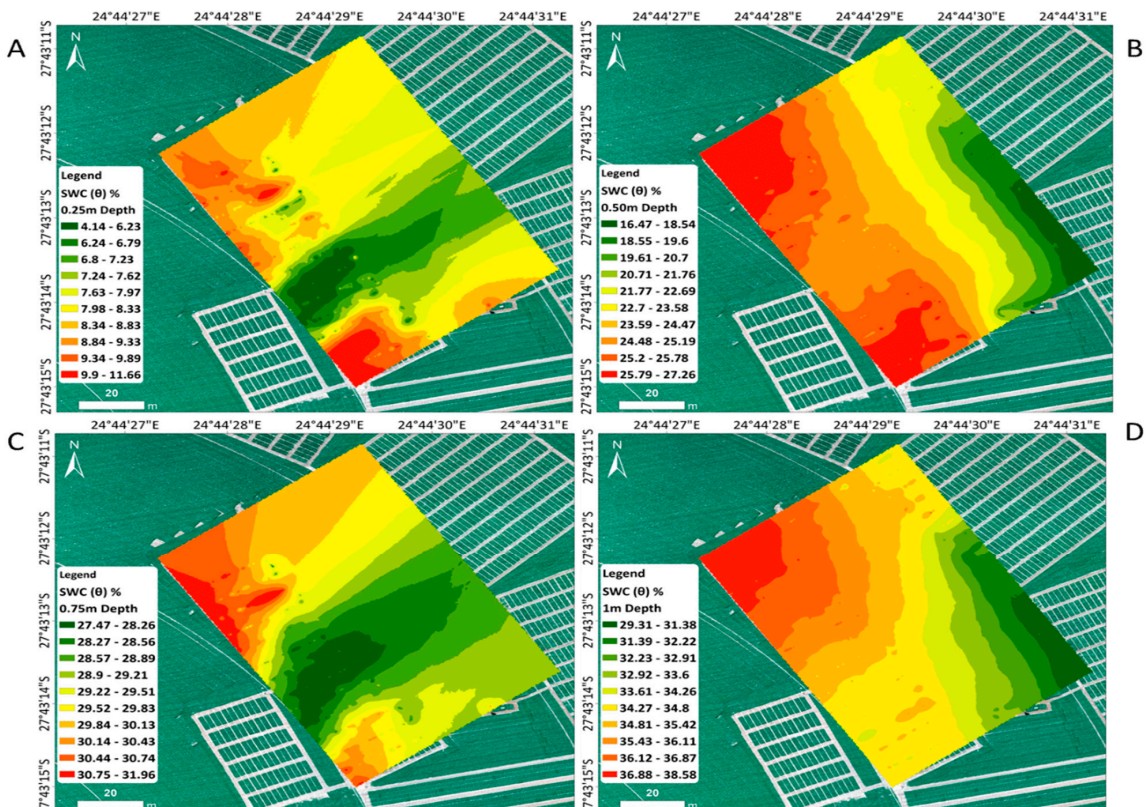

**Figure 13.** Predicted θ at four depths ((**A**–**D**) represents 0.25 m to 1 m on 14 October 2020).

On the following day, 23 September 2020, the soil water content predicted ranged between 5.26 and 39.85 percent, which increased from around the upper soil layer with depth, up to the bottom layer. On the last campaign, conducted on 14 October 2020, the predicted soil water content ranged between 4.34 and 38.58 percent, indicating a rise in soil water content percentage and increasing with depth. An interesting observation is that the bottom layer contains higher soil water content than the top 0.25 m layer, which can be related to evaporation processes and crop water use on the top layer, which is the root zone.

Figure 10 depicts soil water content variability across different soil depths, undertaken during the 8 August 2020 survey. Soil water content during this survey ranged between 4% and 28%. The top layer at 0.25 m depth showed very low soil water content across almost the entire survey area, which can be attributed to soil drying up as it moves away from the irrigation date. An interesting observation is that the bottom layer contains higher soil water content than the bottom layer, which can be related to evaporation processes and crop water use on the top layer, which is the root zone.

Figure 11 shows soil water content spatial distribution maps over four depth intervals, related to the survey done on 22 September 2020. Figure 12A shows the spatial distribution of soil water content between 0 and 0.25 m depth, with high moisture being observed on the western top and bottom of the map. The rest of the layers also indicate a shift of high moisture towards the left side of the surveyed area. The bottom layer contains high moisture zones compared to the rest of the layers, with soil water content ranging between 23% and 39.8%.

Figure 12 portrays the spatial distribution of soil water content across various soil depths up to 1 m. Low soil water content is evident on the right hand sections of the surveyed area on the top two layers (0–0.25 m and 0.50 m), which can be attributed to the active response of the layers relating to soil water evaporation on the top layer and crop water use occurring on the two layers containing the crop roots. The lower layer contains

high soil water content ranging between 16% and 27%. This is lower when compared to the same layer on other survey dates, which can be attributed to time gap between the survey date and the last day of irrigation in the section.

Figure 13 shows the spatial distribution maps of soil water content under different soil depths. It is evident that soil water content on the top layer contains the lowest soil water content range, while the bottom layer contains the highest soil water content. The variability in soil water content can be attributed to the exposure and shielding of soil surface by crop canopy. The upper layer at 0.25 m depth is closer to the surface where the crop density increases as the leaf area and biomass increases, casting protection on the soil from fast evaporation of SWC, where water is only lost through transpiration [59]. In soils exposed to the atmosphere, water evaporates quickly due to the atmospheric demands, while soils shielded by the crop canopy retain moisture for longer periods [59]. The top layers of the surveyed depths contain lower soil water compared to the bottom two layers because they are the root zone levels where crops uptake their water.

3.2.1. Relationship between the Soil Water Content Measured and Predicted Electrical Conductivity

The values extracted from the same geographic location of measurement as the soil water content and the predicted electrical conductivity were used to determine the correlation between the two variables, using linear relationships (Figure 14a–d). During August 2020, the relationship between *ECa* and SWC was negative, with a coefficient of determination $R^2$ of 0.85 and a correlation coefficient of $-0.92$ (Table 2). On 22 September 2020, the relationship between *ECa* and SWC was negative, with an $R^2$ value of 0.95 and a correlation coefficient of $-0.98$. On 23 September 2020 (Table 2), the relationship between *ECa* and SWC was positive, with an $R^2$ value of 0.71 and a correlation coefficient of 0.84 (Table 2). Altdorff et al. [41] claim that a number of variables that vary from region to region, including soil treatments, regulate the relationship between *ECa* and SWC. The silica fertigation, which was applied at a depth of 3 mm on 23 September 2020, may be the reason for the shift from a negative association between *ECa* and SWC to a positive relationship. This may have affected soil conductivity and moisture regulation. The soil's electrical conductivity increases as a result of silica absorbing cations from the soil medium, particularly Ca, mg, and K. These cations function as carriers of electrical current in the soil [60]. Silica was applied to increase the strength of the barley by thickening the cell walls, with the aim of preventing lodging [61]. On 14 October 2020, the relationship between *ECa* and SWC was positive, with an $R^2$ value of 0.75 and a correlation coefficient of 0.87 (Table 2). For all the survey dates, the relationship between *ECa* and SWC portrayed a statistically significant difference $p \leq 0.05$ (Table 2). Therefore, the results show that high soil electrical conductivity was a function of soil water content.

**Table 2.** Statistical summary of the relationship between the measured soil water content and estimated electrical conductivity.

| Day of the Year | Model | r | *p*-Value |
|---|---|---|---|
| **11 August 2020** | $\theta = -3.93\ ECa + 12.54$ | $-0.92$ | 0.04 |
| **22 September 2020** | $\theta = -3.10\ ECa + 39.83$ | $-0.98$ | 0.02 |
| **23 September 2020** | $\theta = 1.09\ ECa - 10.91$ | 0.84 | 0.05 |
| **14 October 2020** | $\theta = 0.67\ ECa + 3.99$ | 0.87 | 0.04 |

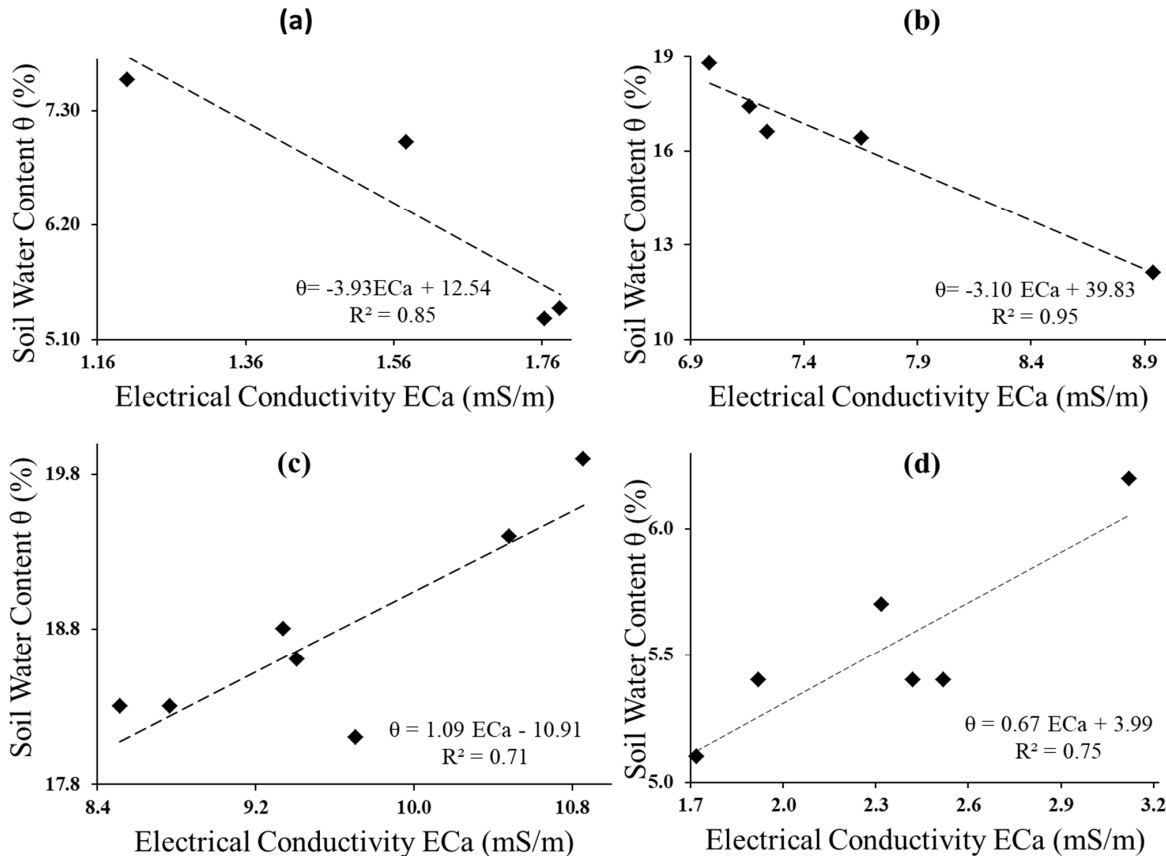

**Figure 14.** The relationship between predicted soil electrical conductivity and measured soil water content, where (**a–d**) represents (11 August 2020, 22 September 2020, 23 September 2020, and 14 October 2020) dates of the survey.

3.2.2. The Relationship between Soil Electrical Conductivity and Soil Physiochemical Properties

A correlation matrix was developed to evaluate the influence of soil properties on electrical conductivity (*ECa*) and (ECe) readings (Table 3). The table shows a very low correlation between ECe, magnesium (mg), and calcium (Ca), with higher positive correlation between ECe and sodium (Na), and a moderate correlation between ECe and potassium (K). Soil pH and ECe showed a relatively low correlation. The correlation matrix below depicts how physiochemical properties of soil relate to the *ECa* measured by the EM38 and an ECe paste of sampled soil with the purpose of obtaining properties, which have higher influence in soil *ECa* variability.

**Table 3.** Correlation matrix of soil physiochemical properties and extractable cations with electrical conductivity *ECa* and ECe.

|     | ECe | Na | K | Mg | Ca | PH | SWC | *ECa* |
|-----|-----|-----|-----|-----|-----|-----|-----|-----|
| **ECe** | 1.00 | | | | | | | |
| **Na** | 0.88 | 1.00 | | | | | | |
| **K** | 0.49 | 0.77 | 1.00 | | | | | |
| **Mg** | 0.49 | 0.52 | 0.43 | 1.00 | | | | |
| **Ca** | 0.05 | 0.41 | 0.75 | 0.03 | 1.00 | | | |
| **PH** | 0.09 | −0.13 | −0.31 | −0.01 | −0.28 | 1.00 | | |
| **SWC** | −0.15 | −0.02 | 0.03 | −0.16 | 0.18 | 0.51 | 1.00 | |
| **Eca** | 0.54 | 0.05 | 0.07 | −0.56 | 0.37 | −0.08 | −0.13 | 1.00 |

### 3.3. Irrigation Events

Figure 15 shows soil water content throughout the barley cropping season, obtained from a DFM soil water content probe. Green arrows indicate the gain in SWC%, which indicates irrigation events. Conversely, a decline in gradient between irrigation events depicts soil water content drawn down through crop water use, indicated by the red line relapse. Soil water content in the figure below was measured between 0–0.5 m depth, which was the active layer and highly responsive to irrigation and crop water use.

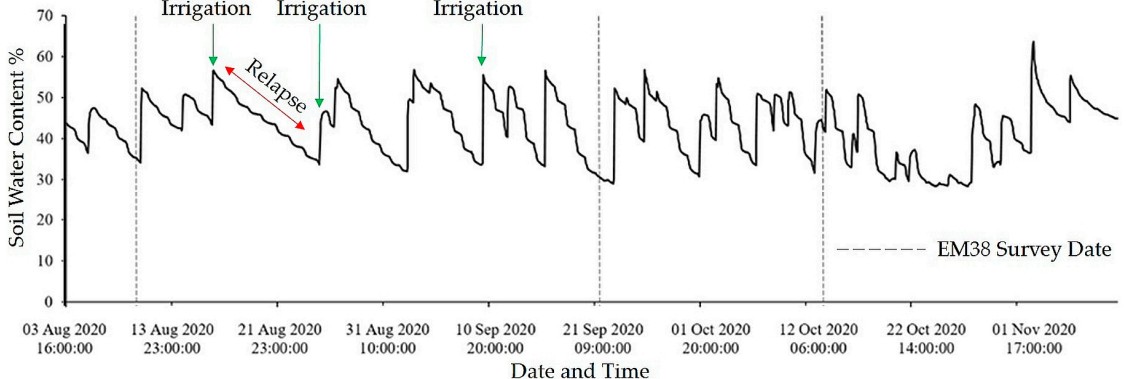

**Figure 15.** Soil water content depicting irrigation events.

The quantity of water given to the crop before each EM38 survey is given in Table 4, including the date and time of each survey campaign. On most irrigation dates, 10 mm drops were given to crops while a 3 mm amount was given during fertigation. Higher *ECa* and SWC were obtained during survey days closer to the irrigation events, while low *ECa* and SWC values were obtained on days further away from the last days of irrigation.

**Table 4.** Irrigation and EM38 survey dates with time, including the amount of water irrigated throughout the season.

| Irrigation Date and Time | Survey Date and Time | Water Irrigated | Condition |
|---|---|---|---|
| **5 August 2020/14:00** | 11 August 2020/15:00 | 10 mm | Dry |
| **21 September 2020/08:00** | 22 September 2020/15:00 | 10 mm | Wet |
| **23 September 2020/12:00** | 23 September 2020/16:00 | 3 mm | Wet (Fertigation) |
| **12 October 2020/21:00** | 14 October 2020/16:00 | 10 mm | Wet-Dry |

### 3.4. Irrigation Water and Soil Quality

Table 5 shows the water quality analysis for the irrigation water used in the farm from the canal. The EC value, magnesium, sodium, calcium, and potassium were analyzed. The sodium absorption ratio (SAR) value obtained from the irrigation water is also reported here. The purpose of this analysis was to compare the EC values between irrigation water and soil *ECa*. From the soil analysis, the SAR average value of soils in the study area was found to be 0.34 mileq/L.

**Table 5.** Irrigation water quality analysis result.

| Measured Parameter Unit | EC mS/m | Ca mg/L | Mg mg/L | Na mg/L | K mg/L | SAR mileq/L |
|---|---|---|---|---|---|---|
| **Value** | 66.6 | 44 | 22 | 51 | 10.91 | 1.108 |

## 4. Discussion

This study focused on assessing the applicability of a non-destructive soil sensing electromagnetic induction device (EM38-MK) in determining the soil's apparent electrical conductivity, with the aim of using it as a proxy to determine soil water content variability post irrigation events, and salinity state of the soil in the Vaalharts irrigation scheme at farm level. To achieve spatial distribution maps of the soil's electrical conductivity at different depths, the geospatial inverse distance weighting interpolation method was used. The model developed based on the relationship between soil water content and soil moisture was used to convert soil electrical conductivity layers into soil water content.

Agricultural soils have different structures, nutritional composition, water holding capacities, and strengths. Electrical conductivity laboratory and field experiments were carried out to investigate their relationship with other soil properties known to have influence on them. Linear regression analysis was conducted between the measured and predicted values of both soil water content and the soil's electrical conductivity to evaluate the capabilities of the EM38 in determining the conductivity of the area and soil water content based on statistical measures. The main influencing factor of high soil *ECa* in this study was SWC%. This is supported by the laboratory tests relating *ECa* and physiochemical properties, which showed fewer correlations, whereas the correlation of *ECa* and SWC was high, with $R^2$ values ranging between 0.71 and 0.95 during different wetness conditions, provided in Figure 8. Similar findings have been reported in some studies [62,63]. These findings clearly demonstrate that as the number of days increase, starting from the days of irrigation, SWC decreases, while *ECa* also has the same pattern.

Regression based analysis between the apparent electrical conductivity and soil water content in this study shows higher correlations with an $R^2 = 0.87$. This illustrates the ability of the EM38-based *ECa* measurements in estimating the soil water content of a given area. Wong and Asseng [63] reported similar correlations with an $R^2$ of 0.78 between *ECa* and available water for plants, while Reedy and Scanlon [62] determined the volumetric water content with a linear regression based model obtaining an $R^2$ of 0.80. Both are not far from what is demonstrated in the findings of this study. These findings are further supported by the work of Sherlock and McDonnell [64] who, upon using the EM38 and EM31 in water table determination, found that simple linear regression explained more than 80% of variations. High correlation coefficients (r) were obtained between measured soil water content and estimated *ECa* ($-0.92$, $-0.98$, 0.84, and 0.87) making it evident that the relationship between *ECa* and soil water content can be used as a surrogate in order to estimate one of the two soil variables. High *ECa* values can be a sign of salinity according to Bai et al. [57], however, based on our findings this was not the case. Furthermore, the relationship between *ECa* and the physiochemical properties were low, leaving SWC as the main driver of *ECa*.

The findings of this study demonstrate that the soil water content is the main driver that allows the flow of electrical current in the soil medium, making it easier for the electromagnetic induction device to determine the soil's apparent electrical conductivity. Such findings are also reported by others [36,43,65]. The linear relationship between soil water content and the soil's electrical conductivity made it possible to easily convert *ECa* (mS/m) into SWC%, influenced largely by irrigation in the area. As the days got further away from the irrigation days, the decline in soil electrical conductivity was evident. This was based on the fact that soil water content declines as the crop uses water and evaporation takes place in the absence of precipitation, which makes it easier for farmers to determine and track their soil water depletion. This provides an accurate basis for irrigation scheduling and quantifying crop water use.

Misra and Padhi [66], when using the EM38 for soil water distributions, found that the best time to use the EM38 for soil water content mapping was when the *ECa* was highly influenced by soil water content. In this study, we found that with the sandy-loam soils in the arid environment where the study was done, *ECa* can only determine soil water content under the equality: *ECa* > 0 < 113 mS/m. For that reason, *ECa* cannot determine SWC if

it is less than or equal to zero, or if it exceeds 113 mS/m as per the limits of the model developed in this study.

On assessing the relationship between *ECa* and other soil parameters, the findings demonstrated that magnesium (mg) correlated slightly above average with *ECa*, while other physiochemical properties showed very low correlation, which provides information that *ECa* in our study area is not directly influenced by other physiochemical properties of the soil apart from soil water content. These observations provide evidence that the EM38-based *ECa* measurement can accurately determine SWC in the area. Kurt Heil and Schmidh [67], on assessing the influence of soil properties on *ECa*, also found similar results on the basis that the relationship between *ECa* and soil properties was very weak in their surveys, making soil water content the main driver of *ECa*.

The SAR average value of soils in the study area was found to be 0.34 mileq/L, which is low sodium when SAR ranges between 0–10. As such, it indicates a low risk of irrigation water being used as described by Burger and Celkova [68]. Results from irrigation water quality analysis showed a higher value in SAR with 1.11 mileq/L, which is higher than the value obtained in the soil, however, although the SAR value in water is higher than the one in soil, it still falls within the safe irrigation water boundary according to Burger and Celkova [68]. The EC obtained in water is also higher than the one obtained from soil, which is still within the safe saline levels according to another study [56].

Furthermore, on analyzing the spatial distribution maps of *ECa*, we compared the observed, predicted, and local salinity scale values of South Africa found in one study [56] to assess if there was any evidence of salinity in the area as reported in its history. We found that soils within the study area had no sign of salinity, with *ECa* values < 100 mSm$^{-1}$. The findings of this study relate to the findings of other studies [43,65], which also used the EM38-mk to estimate soil moisture based on *ECa* measurements, and found similar trends in terms of correlations and behavior of *ECa* under given SWC conditions.

The use of a non-destructive electromagnetic induction device has the potential to reduce the time and costs of installing point sensors in the field or avoid the necessity of making actual measurements using portable sensors, particularly in solving soil water related problems in agriculture, hydrogeological environments, and the natural environment as a system [43]. The capability of the EM38 to map soil parameters with depth that covers most root zones of common crops that are grown around the world offers the opportunity to solve the spatial limitations of point fixed data [43]. In this study, the findings demonstrate that SWC in the top layer is always lower when compared to the bottom layers, which can be related to the fact that the top 0–0.25 m soil layer is occupied by roots, although some crops can go deeper with growth stages.

The spatial variability of soil electrical conductivity and soil water content was successfully determined, and the generated maps come with a number of benefits. These findings can be used to assist with scheduling irrigation for irrigation technicians, who would be able to understand the soil water content variability beyond the naked eye on the surface and eliminate the issue of providing information using spatial disjoined figures. The produced maps in precision agriculture, where spot soil treatment is done using RTK-linked auto-machineries, could be the used as base maps to treat only areas which are necessary, saving the costs of soil treatment resources and labor costs.

Although the EM38 is capable of mapping soil *ECa* and SWC, there are limitations of such devices; in this study the existence of the metallic pivot irrigation tower, fencing, metallic storage facilities, and metal pegs in the field restricted the area of survey. The EM38 devices can only be used for limited periods unless they are vehicle towed. Each soil type requires local calibrations when the EM38 is used as a proxy to determine soil water content because the EM38 does not directly determine soil water content. The soil's electrical conductivity is the main component that is measured, and can be influenced by different physiochemical properties [69]. It was found that the model can predict soil water content based on the soil's apparent soil electrical conductivity if *ECa* > 0 < 113 mS/m for our study area.

The model developed in this study was based on a small section of the farm (100 m by 70 m) to avoid the effects of metallic objects during the electromagnetic survey. For that reason, not many samples were taken from the selected area. The number of samples, however, does not have an impact on the performance of our model in retrieving SWC from *ECa*; a similar case has been reported in one study [70], although they used linear models. Future studies are recommended to incorporate more samples and cover an extensive area without signal obstacles to assess the model performance. With the benefits of localized scales, this study recommends the use of EM38 surveys in conjunction with spatial covering sensors that are capable of estimating soil water content, including the use of UAV-based sensors, which are capable of providing high resolution images and information on soil water content at shallow depths.

## 5. Conclusions

Soil water content variability at different depths under different wetness conditions was successfully estimated using electromagnetic induction as a proxy. The determination of soil salinity using soil electrical conductivity was also successful, finding that the study area soils are below the saline levels of *ECa*. The water analysis results also indicated EC levels that are below the salinity levels. Based on the integrated analysis of different soil physiochemical parameters, it was observed that *ECa* in the study area was more influenced by soil water content. Soil moisture using direct point specific techniques is usually reported using spatial disjoined figures, which do not really provide information that can be used in variable irrigation treatments. The electromagnetic induction instrument (EM38) is worth using to quantify the spatial variability of soil water content by measuring the spatial variability of soil electrical conductivity. A significant observation, in the simulation of soil water content variability based on soil electrical conductivity, was that soil electrical conductivity response to the instrument occurs when there is enough water content available in the soil to move the charges around. Although the results proved that soil electrical conductivity measurement could be used as a surrogate for determining soil water content using EM38, calibration for every soil type is required for better results. On the other hand, the instrument is expensive, and the interpretation of data requires scientific analysts rather than simply people with agricultural knowledge. We recommend future research to look at large field coverage with more point samples. The approaches and methods used in this study can be replicated in other areas to attain an in-depth understanding of the soil water dynamics as well as understanding the salinity status of the soil.

**Author Contributions:** Conceptualization, methodology, analysis, and original draft preparation P.E.R.; writing—review and editing, M.A.M.A.E.; writing—review and editing, E.A.; writing—review and editing, J.G.C.; writing—review and editing, G.L.; review and editing, funding acquisition, M.A.M.A.E. and J.G.C.; fieldwork operations and editing, P.E.R., E.B.E. and M.A.M.A.E. All authors have read and agreed to the published version of the manuscript.

**Funding:** This research was funded by the Agricultural Research Council-Natural Resources and Engineering (ARC-NRE) Project number: P07000128, Department of Science and Innovation; National Research Foundation and the Water Research Commission of South Africa Project number: C2022/2023-00978.

**Data Availability Statement:** Available on request.

**Acknowledgments:** We would like to thank Fagan Scheepers and the entire South African Barley Institute (SABBI) for permitting us to use their experimental farm for this study, we acknowledge the Water Research Commission of South Africa (WRC) for additional support in this work. Finally we would also like to acknowledge the creditable work done by the reviewers and editors at water and Thomas Fyfield for helping with in-house review.

**Conflicts of Interest:** The authors declare no conflict of interest.

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
