# Peer review of "Determination of Soil Electrical Conductivity and Moisture on Different Soil Layers Using Electromagnetic Techniques in Irrigated Arid Environments in South Africa"

_water, doi:10.3390/w15101911_

Round 1

Reviewer 1 Report

Kindly see the attached word document

Author Response

To: The reviewer and editors

We would like to thank you and appreciate you for your efforts and time in reviewing our manuscript and providing constructive inputs to enhance the scientific output of our manuscript. We have attended to all the comments and suggestions provided after your review. Our response are given on the table below.

Reviewer’s comments

Authors’ response

You mention SAR cations - have you determined the SAR value and if so, it would be useful to present it.

Thanks very much for pointing this out. We have included the SAR value on our result and discussion section for both soil

Has the irrigation water been analyzed? if so, it would be good to show the results of the analysis.

Thanks for pointing this, we have included the water quality results on the manuscript.

The irrigation rate (10 mm) is quite low, yet you mentioned that the 0,8 m of soil depth was

influenced by irrigation events. Please clarify this!

We appreciate pointing this out, we had a typing error, the depth is 0-0.25 m which is the root zone and the most responsive layer to irrigation, and we have rectified this on the manuscript.

Kind regards,

Reviewer 2 Report

30 Apr 2023

Reviewer’s queries:

Determination of Soil Electrical Conductivity and Moisture on 2 Different Soil Layers Using Electromagnetic Techniques in Irri-3 gated Arid Environments in South Africa

Line 290-294 Authors have mentioned: “Figure 5 indicates the spatial distribution of soil ECa for four depths in the surveyed area. It is clear that higher ECa values are more pronounced on the western part of the map cutting almost half of the surveyed area while the eastern part has slightly low values which are higher compared to the distribution indicated in Figure 4. This is an indication of high soil water content allowing more conductivity in soils. ECa in this case ranged between 5 and 19.4 mS/m”

Reviewer’s query: Please provide a justification/reference to support the claim that the western part has high soil water content than the eastern part.

Line 298-303 Authors have mentioned: “Figure 6 depicts an increase in soil ECa across all depths layers. ECa ranges between 7.7 and 20.3 mS/m. Higher ECa is evident in the northern and southern parts of the surveyed area while appeared to be lower in the central parts of 0-25 m upper layer and 0.75 m while in 0.50 and 1 m ECa appeared to be higher in the western part of the surveyed area. The higher ECa might be attributed to high soil water content allowing high conductivity in soils”

Reviewer’s query: Please provide a justification/reference to support the claim that “The higher ECa might be attributed to high soil water content allowing high conductivity in soils.”

Line 322-325 Authors have mentioned:“During August 2020, 319 the relationship between ECa and SWC was negative with a coefficient of determination 320 R2 of 0.85, a correlation coefficient of -0.92 (Table 2). On 22 September 2020, the relationship between ECa and SWC was negative, with the R2 value of 0.95 and the correlation coefficient of -0.98. On 23 September 2020 (Table 2), the relationship between ECa and SWC was positive, with an R2 value of 0.71 and a correlation coefficient of 0.84 (Table 2). On 14 October 2020, the relationship between ECa and SWC showed a positive relationship with the R2 value of 0.75 and a correlation coefficient of 0.87 (Table 2).”

Reviewer’s query: Please provide a conceptual understanding why the relationship between ECa and SWC turned from negative to positive in one day.

Figure 10. Model used to convert ECa to soil water content (θ).

Reviewer’s query: Please discuss if there are any limitations to the above model? As it can be seen that there are a few data points that are used to get the given model relationship. Are authors concerned of the limited data points in Figure 10 and on Figures 8, 9. Has that been discussed in the manuscript?

Line 409

variability in soil water content can be attributed to the exposure and shielding of soil surface by crop canopy.

Reviewer’s query: Please provide a justification and support what leads to author’s above finding.

30 Apr 2023

Reviewer’s queries:

Determination of Soil Electrical Conductivity and Moisture on 2 Different Soil Layers Using Electromagnetic Techniques in Irri-3 gated Arid Environments in South Africa

Line 290-294 Authors have mentioned: “Figure 5 indicates the spatial distribution of soil ECa for four depths in the surveyed area. It is clear that higher ECa values are more pronounced on the western part of the map cutting almost half of the surveyed area while the eastern part has slightly low values which are higher compared to the distribution indicated in Figure 4. This is an indication of high soil water content allowing more conductivity in soils. ECa in this case ranged between 5 and 19.4 mS/m”

Reviewer’s query: Please provide a justification/reference to support the claim that the western part has high soil water content than the eastern part.

Line 298-303 Authors have mentioned: “Figure 6 depicts an increase in soil ECa across all depths layers. ECa ranges between 7.7 and 20.3 mS/m. Higher ECa is evident in the northern and southern parts of the surveyed area while appeared to be lower in the central parts of 0-25 m upper layer and 0.75 m while in 0.50 and 1 m ECa appeared to be higher in the western part of the surveyed area. The higher ECa might be attributed to high soil water content allowing high conductivity in soils”

Reviewer’s query: Please provide a justification/reference to support the claim that “The higher ECa might be attributed to high soil water content allowing high conductivity in soils.”

Line 322-325 Authors have mentioned:“During August 2020, 319 the relationship between ECa and SWC was negative with a coefficient of determination 320 Rof 0.85, a correlation coefficient of -0.92 (Table 2). On 22 September 2020, the relationship between ECa and SWC was negative, with the Rvalue of 0.95 and the correlation coefficient of -0.98. On 23 September 2020 (Table 2), the relationship between ECa and SWC was positive, with an Rvalue of 0.71 and a correlation coefficient of 0.84 (Table 2). On 14 October 2020, the relationship between ECa and SWC showed a positive relationship with the Rvalue of 0.75 and a correlation coefficient of 0.87 (Table 2).”

Reviewer’s query: Please provide a conceptual understanding why the relationship between ECa and SWC turned from negative to positive in one day.

Figure 10. Model used to convert ECa to soil water content (θ).

Reviewer’s query: Please discuss if there are any limitations to the above model? As it can be seen that there are a few data points that are used to get the given model relationship. Are authors concerned of the limited data points in Figure 10 and on Figures 8, 9. Has that been discussed in the manuscript?

Line 409

variability in soil water content can be attributed to the exposure and shielding of soil surface by crop canopy.

Reviewer’s query: Please provide a justification and support what leads to author’s above finding.

Author Response

To: The reviewer and editors

We would like to thank you and appreciate you for your efforts and time in reviewing our manuscript and providing constructive inputs to enhance the scientific output of our manuscript. We have attended to all the comments and suggestions provided after your review. Our response are given on the table below.

Reviewer’s comments

Authors’ response

Line 290-294 Authors have mentioned: “Figure 5 indicates the spatial distribution of soil ECa for four depths in the surveyed area. It is clear that higher ECa values are more pronounced on the western part of the map cutting almost half of the surveyed area while the eastern part has slightly low values which are higher compared to the distribution indicated in Figure 4. This is an indication of high soil water content allowing more conductivity in soils. ECa in this case ranged between 5 and 19.4 mS/m”

Reviewer’s query: Please provide a justification/reference to support the claim that the western part has high soil water content than the eastern part.

Thanks for pointing this out, we have added support the claims on the manuscript.

It is evident on the map Figure 4_A mean SWC=6.33% as compared to the Figure 5 where the mean SWC=16.26% has increased while the ECa values also increased with maximum ECa=13.75 mS/m. Similar findings have been described in the works of Bai et al., 2013 and Turkeltaub et al., 2022.

Line 298-303 Authors have mentioned: “Figure 6 depicts an increase in soil ECa across all depths layers. ECa ranges between 7.7 and 20.3 mS/m. Higher ECa is evident in the northern and southern parts of the surveyed area while appeared to be lower in the central parts of 0-25 m upper layer and 0.75 m while in 0.50 and 1 m ECa appeared to be higher in the western part of the surveyed area. The higher ECa might be attributed to high soil water content allowing high conductivity in soils”

Reviewer’s query: Please provide a justification/reference to support the claim that “The higher ECa might be attributed to high soil water content allowing high conductivity in soils.”

Thanks for pointing this out, we have added support the claims on the manuscript.

This is due to the fact that the mean θ% continues to increase when comparing figure 5 where mean θ is 16.26% and figure 6 is 18.77%  while the ECa values on the maps also in-crease, the relationship where ECa increases with SWC has been narrated by Bai et al., 2013.

Line 322-325 Authors have mentioned: “During August 2020, 319 the relationship between ECa and SWC was negative with a coefficient of determination 320 R2 of 0.85, a correlation coefficient of -0.92 (Table 2). On 22 September 2020, the relationship between ECa and SWC was negative, with the R2 value of 0.95 and the correlation coefficient of -0.98. On 23 September 2020 (Table 2), the relationship between ECa and SWC was positive, with an R2 value of 0.71 and a correlation coefficient of 0.84 (Table 2). On 14 October 2020, the relationship between ECa and SWC showed a positive relationship with the R2 value of 0.75 and a correlation coefficient of 0.87 (Table 2).”

Reviewer’s query: Please provide a conceptual understanding why the relationship between ECa and SWC turned from negative to positive in one day.

Thanks for pointing this out, we have provided support to the changes that occurred in one day aligned with the fertigation event that took place at 3 mm.

Altdorff et al. (2018) claim that a number of variables that vary from region to region, including soil treatments, regulate the relationship between ECa and SWC. The silica fertigation, which was applied at a depth of 3 mm on 23 September 2020, may be the reason for the shift from a negative association between ECa and SWC to a positive relationship. This may have affected soil conductivity and moisture regulation. The soil's electrical conductivity increases as a result of silica absorbing cations from the soil medium, particularly Ca, mg, and K, these cations function as carriers of electrical current in the soil (Darwesh et al., 2019).  Silica was applied to increase the strength of barley by thickening the cell walls with the aim of preventing lodging (Guerriero et al., 2016).

Figure 10. Model used to convert ECa to soil water content (θ).

Reviewer’s query: Please discuss if there are any limitations to the above model? As it can be seen that there are a few data points that are used to get the given model relationship. Are authors concerned of the limited data points in Figure 10 and on Figures 8, 9. Has that been discussed in the manuscript?

Thanks for pointing this out, we have added this on our discussions:

The model developed in this study was based on a small section of the farm (100 m X 70 m) to avoid the effects of metallic objects during the electromagnetic survey, for that reason not many samples were taken from the selected area. The number of samples however does not have an impact on the performance of our model to retrieve SWC from ECa, a similar case has been demonstrated in the study of (Foley and Boulton, 2015) although they used linear models.

Line 409

variability in soil water content can be attributed to the exposure and shielding of soil surface by crop canopy.

Reviewer’s query: Please provide a justification and support what leads to author’s above finding.

Thanks for pointing this out, we have added this on our manuscript:

The upper layer 0.25 m depth is closer to the surface where the crop density increases as the leaf area and biomass increases, casting protection on the soil from fast evaporation of SWC where water is only lost through transpiration (Wallace et al., 2018). Soils exposed to the atmosphere evaporates water fast due to the atmospheric demands while soils shielded by crop canopy contains moisture for longer periods (Wallace et al., 2018).

Kind regards,

Eugene 07 May 2023

Reviewer 3 Report

The article “Determination of Soil Electrical Conductivity and Moisture on Different Soil Layers Using Electromagnetic Techniques in Irrigated Arid Environments in South Africa” present the evaluation of an EM38-MK equipment as a non-destructive electromagnetic induction device to mapping the distribution of soil electrical conductivity and soil moisture at different soil depths.

The manuscript needs some improvement before further processing. My recommendations are below:

-          The introduction is complete but difficult and hard to read. Please include different paragraphs, all the information is in the same very long paragraph.

-          The aim of the article are included in the Introduction section but please, include the novelty of the article in the abstract and in the introduction section. Which are the initial hypothesis of the article?

-          Please introduce the meaning of A, B and C in figure1. Lines 163-164.

-          Please, correct the coordinates (line 141)

-          Please include a, b and C in figure 2

-          Rewrite discussion and results sections. There are several paragraph without references but in the discussion section a comparison of the results should be done. For example, information of lines 434-440 is not a discussion is an introduction and there are not references. The information of lines 454-458 is not also a discussion; this information should be in methods. The information of lines 459-473 is an explanation of the results, not a discussion. There are more examples, please rewrite the discussion and results sections.

Author Response

To: The reviewer and editors

We would like to thank you and appreciate you for your efforts and time in reviewing our manuscript and providing constructive inputs to enhance the scientific output of our manuscript. We have attended to all the comments and suggestions provided after your review. Our response are given on the table below.

Reviewer’s comments

Authors’ response

The introduction is complete but difficult and hard to read. Please include different paragraphs, all the information is in the same very long paragraph.

Thank you for pointing this out, we have changed the introduction into section paragraphs.

The aim of the article are included in the Introduction section but please, include the novelty of the article in the abstract and in the introduction section. Which are the initial hypothesis of the article?

The novelty of the article has been included in the introduction and abstract. The hypothesis of the article is that electromagnetic induction device (EM38-MK) can be used to estimate soil water content and salinity using electrical conductivity as a surrogate which can be integrated in geospatial models to provide spatial variability.

Please introduce the meaning of A, B and C in figure1. Lines 163-164.

Section fixed:

Figure 1. Locality map of the study area showing the experimental farm and the area of interest used for the EM38 survey where (a) is the 18 ha experimental farm, the red boundary is the EM38 surveyed area, (b) shows the provincial layout of South Africa highlighting Northern Cape (NC) where the study area is located in green while (c) shows the location of the survey area in the district municipality.

Please, correct the coordinates (line 141)

Coordinates fixed:

Geographically, the 18-ha experimental farm is located between the coordinates: -27°43'14.41"S and 24°44'34.18"E.

Please include a, b and C in figure 2

Included:

Figure 2. An electromagnetic induction instrument (EM38MK) and its components, where (a) is the EM38 device, (b) is the device on vertical mode during survey, (c) shows the setting controls and (d) is the Bluetooth GPS linked to the device for sample location

Rewrite discussion and results sections. There are several paragraph without references but in the discussion section a comparison of the results should be done. For example, information of lines 434-440 is not a discussion is an introduction and there are not references. The information of lines 454-458 is not also a discussion; this information should be in methods. The information of lines 459-473 is an explanation of the results, not a discussion. There are more examples, please rewrite the discussion and results sections.

Thanks for pointing this out.

The result and discussion has been re-written with the necessary references cited.

Kind regards,

Eugene 07 May 2023

Round 2

Reviewer 2 Report

Authors have responsed to the previous queries satisfactorily.

Reviewer 3 Report

It is an improved version of the article because reviews have been taken into account. However, there is an error in the references, they cannot be displayed in the article.